# Understanding COVID-19-Related Behaviors, Worries, and Attitudes among Chinese: Roles of Personality and Severity

**DOI:** 10.3390/bs14060482

**Published:** 2024-06-06

**Authors:** Jie Liu, Chun Cao, Yanyan Zhang

**Affiliations:** 1Faculty of Education, Northeast Normal University, Changchun 130024, China; liujie@nenu.edu.cn (J.L.); caogecheng@aliyun.com (C.C.); 2School of Philosophy and Sociology, Jilin University, Changchun 130012, China

**Keywords:** COVID-19, severity, personality, behavioral responses, worries, attitudinal responses

## Abstract

During the COVID-19 pandemic, people exhibited various forms of adjustments. This study examines how situational factors (i.e., the severity of COVID-19) and individual differences (i.e., the HEXACO traits) affect one’s COVID-19-related responses regarding behaviors (i.e., mask-wearing and hoarding), worries (i.e., worrying about infecting and spreading COVID-19), and attitudes (i.e., discrimination and empathy toward people infecting COVID-19) in China. With a sample of 927 participants, our results show that the severity of COVID-19 was predictive of all the responses, and its predictive value was more pronounced relative to personality traits. Concerning the association between personality traits and responses, Honesty-Humility and Conscientiousness were predictive of one’s behaviors, Emotionality was predictive of one’s worries, and almost all the HEXACO traits were associated with one’s attitudes toward people infected with COVID-19. This study sheds some light on understanding how situations and individual differences shape one’s responses in a time of emergency.

## 1. Introduction

The coronavirus disease 2019 (COVID-19) has greatly impacted people’s lives globally. Indeed, after the World Health Organization (WHO) declared COVID-19 a worldwide pandemic and called on countries to take urgent action on 11 March 2020, so far, around 120 million people have been affected by COVID-19, and more than 2 million people have died because of it [1]. Given the severity of COVID-19, some countries have enforced policies to curb its spread, such as lockdowns, travel restrictions, school closures, and social distancing [2].

Enacting these rules has been shown to be effective in reducing both the individual risks of infection and the likelihood of contracting others [3,4]. However, people differ in the extent to which they follow these rules. Some people may strictly follow the recommendations, whereas others may resist them. Since individual differences in behaviors, thoughts, and feelings are well represented by personality [5], one’s personality is likely to be predictive of one’s responses during the COVID-19 pandemic.

Indeed, recent research has reported some associations between personality and one’s COVID-19-related responses [6,7,8]. However, very few studies have jointly examined the effects of both individual differences and situational factors on people’s COVID-19-related responses. This study aims to fill the gap by investigating how personality and the severity of COVID-19 affect one’s COVID-19-related responses, including behaviors, worries, and attitudes, in a Chinese sample.

### 1.1. Personality and COVID-19-Related Responses

Some studies have already investigated the link between personality traits and people’s COVID-19-related responses, especially concerning behaviors like mask-wearing and hoarding, since the outbreak of COVID-19. However, the results are quite mixed. Concerning mask-wearing, Zettler et al. [8] reported that both Honesty-Humility and Conscientiousness were positively correlated with complying with the recommendations and rules to fight COVID-19 in Danish and German samples. By contrast, Shook et al. [9] reported that higher Conscientiousness and Openness to Experience were related to less use of face masks in a US sample. Barceló and Sheen [10] found that introverted people are more likely to resist wearing masks in Spain.

Another widely examined behavior is hoarding. Garbe et al. [7] found that people high in Conscientiousness are more likely to stockpile toilet paper in samples from the US, Canada, or Europe. Dammeyer [11] reported that stockpiling was positively associated with Extraversion and Neuroticism while negatively associated with Conscientiousness and Openness to Experience in samples from Denmark and British. Columbus [12] reported a negative association between Honesty-Humility and hoarding behaviors in a British sample, whereas Garbe et al. [7] found no support for such a link. The mixed results of these previous studies highlight the importance of having more research on this issue.

### 1.2. Situational Factors and COVID-19 Related Responses

Besides personality, some situational factors also impact individuals’ responses during the COVID-19 pandemic. For example, recommendations from authorities can increase people’s mask-wearing behavior [13]. Also, individuals experience more negative and less positive emotions if their residence is more severely affected by COVID-19 [14]. Besides this, when conspiracy belief about COVID-19 (e.g., COVID-19 was created as a biological weapon) is predominant, people resist taking preventive behaviors [15].

Like the previously mentioned situational factors, the severity of COVID-19 is also likely to influence people’s responses. People perceiving COVID-19 as severer should be more likely to engage in protective behaviors. Indeed, studies examining the relationship between the severity of COVID-19 and one’s vaccination intention have reported a positive correlation, indicating that the severer COVID-19 is, the more likely people are to take preventive actions [16,17]. Accordingly, it is highly possible that the severity of COVID-19 is predictive of one’s responses during the pandemic.

### 1.3. The Current Study

Noticeably, both personal factors and situational factors could play a role in shaping one’s responses. Decades ago, Kurt Lewin [18] noted that a person’s behavior (B) is a function of the person (P) and her/his environment (E), expressed as B = *f*(P, E). This famous statement has triggered great debate among later researchers in determining the relative predictive power of personal factors and situational factors in behaviors, emotions and attitudes [19,20,21]. Relating to the current study, it is intriguing to examine the relative predictive power of person factors, represented by personality traits, and environmental factors, represented by severity of COVID-19, in relation to COVID-19-related responses.

Besides the pre-mentioned COVID-19-related responses, such as mask-wearing and hoarding, there are other relevant reactions related to the COVID-19 pandemic that need to be further examined. Worries are frequently mentioned during the COVID-19 pandemic, and people tend to worry about becoming infected with COVID-19 themselves or transmitting COVID-19 to others [16,22]. Apart from being concerned about oneself during the COVID-19 pandemic, people also have different attitudes, such as empathy and discrimination, toward those who are vulnerable to coronavirus. Empathy describes one’s concern for and understanding of another person’s experience [23]. During COVID-19, people with more empathetic concerns toward those who are vulnerable to coronavirus were more likely to adhere to the recommendations [24]. Discrimination is conceptualized as harmful actions toward people because of their membership in a particular group [25]. During the COVID-19 pandemic, people who had been infected by coronavirus, even though they had recovered from that illness, were more likely to be discriminated against [26].

Some studies have already illustrated that personality traits are associated with one’s tendency to experience anxiety [27], empathetic concern to others [28], and prejudice toward derogated groups [29]. However, few studies have directly investigated the relations between personality traits and these COVID-19-related responses. Besides this, almost no studies have examined the effects of personality traits and severity of COVID-19 jointly. To shed some light on this issue, this study explores how personality traits (i.e., person factor) and the severity of COVID-19 (i.e., situation factor) affects one’s COVID-19-related responses, including behaviors (i.e., hoarding and mask-wearing), worries (i.e., worry about contracting and spreading COVID-19), and attitudes (i.e., empathy and discrimination toward people infected with COVID-19), in a Chinese sample.

## 2. Method

### 2.1. Participants

We collected data in the midst of the second outbreak of COVID-19 between 28 January and 7 February 2021 in China. Participants were recruited via online advertisement and social media; 1132 participants started our survey, and 927 (18% dropout rate) completed it. Their age ranged from 18 to 67 (*M* = 30.79, *SD* = 10.46). Most of them were female (76%), had a job (51%), married or in a relationship (62%), and had at least a bachelor’s degree (89%).

### 2.2. Measures

Personality was measured with the Brief HEXACO Inventory (BHI) [30], comprising 24 items and measuring each personality trait with 4 items. Sample items include “I find it difficult to lie” (Honesty-Humility) and “I worry less than others” (Emotionality reverse-scored). These questions were answered with a 5-point Likert scale (1 = strongly disagree, 5 = strongly agree).

Three items measured the severity of COVID-19. “COVID-19 is a severe disease” and “How likely do you think it is that a person who falls ill with COVID-19 dies as a result of the disease” were selected from Karlsson et al. [16]. “COVID-19 is a very infectious disease” was included to capture the high-contagiousness of COVID-19. These questions (including all the scales below) were answered with a 7-point Likert scale (1 = strongly disagree, 7 = strongly agree).

COVID-19-related behaviors include mask wearing and hoarding. Mask wearing was measured with one item, “I am willing to wear a mask when going outside in response to coronavirus”, and hoarding was measured with one item, “In response to coronavirus, I have bought more food or supplies than usual”.

COVID-19-related worries include worry about contracting and spreading coronavirus. Worry about contracting coronavirus was measured with three items, “How likely do you think that you will be infected with coronavirus?”, “How much do you worry about infecting coronavirus?” and “How severe would it be for your health if you contracted coronavirus?” These items were adapted from Karlsson et al. [16]. Worry about spreading coronavirus was measured by two items, “How much do you worry about transmitting coronavirus to someone else?” and “How guilty would it be for you if you transmit coronavirus to someone else?”.

COVID-19-related attitudes include empathy toward people who had coronavirus and discrimination toward people who had recovered from coronavirus. Empathy was measured by two items, selected and adapted from the empathy scale [31], including “I feel empathy for the patients infected with coronavirus” and “I feel tenderness for the patients infected with coronavirus”. Discrimination was measured by two items, selected and adapted from scales for measuring fear of AIDS and homophobia [32], including “I wouldn’t mind being in the same room with a friend who had recovered from coronavirus” and “If I found out a friend had coronavirus before, I would be afraid to hug him/her”.

## 3. Results

Table 1 displays the descriptive statistics, Pearson correlations, and reliabilities (i.e., Cronbach *α*) of all variables involved in the study. The severity of COVID-19 was positively correlated with all COVID-19-related responses. The correlation between personality traits and COVID-19-related responses was less clear, but in general, each HEXACO trait correlated with at least two outcomes.

Table 2 presents the results of multiple regressions predicting COVID-19-related responses from demographics (i.e., age and gender), the HEXACO traits, and the severity of COVID-19. For each outcome, we constructed three hierarchical regression models, with demographic variables set as predictors in Step 1, then adding personality traits to these models in Step 2, and finally adding the severity of COVID-19 in Step 3. In this way, we can dissect the amount of variance in the outcome variables explained by each group of predictors.

Adding the HEXACO traits into the regression models significantly improved the model fit for all outcomes (4.90 ≤ *F* ≤ 13.19, all *p* < 0.01) except hoarding (*F* = 1.49, *p* = 0.180), accounting for more incremental variance (ranging from 3% to 7%) compared to models with only age and gender as predictors. Concerning COVID-19-related behaviors, mask-wearing was predicted by higher levels of Honesty-Humility and Consciousness, and hoarding was predicted by higher levels of Emotionality. Concerning COVID-19-related worries, worrying about contracting coronavirus was predicted by higher levels of Emotionality, and worrying about spreading coronavirus was predicted by higher levels of Honesty-Humility, Emotionality, Extraversion, and Agreeableness. Concerning COVID-19-related attitudes, discrimination was predicted by lower levels of Honesty-Humility, Agreeableness, and Openness to Experience, and higher levels of Emotionality and Consciousness. Empathy was predicted by all the HEXACO traits, with people higher on the six traits feeling more empathetic toward individuals infected with coronavirus.

Adding the severity of COVID-19 in the previous models greatly improved the model fit for all the outcomes (8.52 ≤ *F* ≤ 226.49, all *p* < 0.01) and accounted for more incremental variance (ranging from 1% to 20%) compared to models with only demographics and personality traits. Generally, the severity of COVID-19 was positively associated with all COVID-19-related responses, with coefficients ranging from 0.09 to 0.44.

## 4. Discussion

This study examined the effects of situational factors (i.e., the severity of COVID-19) and individual differences (i.e., the HEXACO personality traits) on one’s COVID-19-related responses (i.e., behaviors, worries and attitudes) during the outbreak of COVID-19 in China. We found that both factors were predictive of one’s behaviors, worries, and attitudes during the pandemic, and the severity of COVID-19 has higher predictive power toward these outcomes relative to personality traits. In general, our study provides further support to the formula B = *f*(P, E) by demonstrating that both personality traits and situations play a role in shaping one’s responses during a period of uncertainty and turbulence. Concerning personality, our study provides support to the previous conclusion that personality traits are predictive of a variety of life outcomes in general [33,34], and extend these conclusions by demonstrating the predictive power of personality traits on one’s experiences and actions in times of uncertainty and ambiguity. Concerning environment, our study shows that perceiving coronavirus as a severe illness is associated with more strategic adaptions and compliance with commendations even after controlling for one’s personality traits. This implies that situational factors tend to outweigh individual differences in predicting one’s responses during an emergency. This implication nicely mirrors the previous literature emphasizing the importance of the situation, especially when informational cues are unambiguous, behavioral expectations are clear, there are incentives to comply, and people can meet the behavioral demand [35,36]. Additionally, the relatively higher predictive power of the severity of COVID-19, compared to personality traits, implies that environmental factors are more likely to stand out when people are in an unexpected emergency.

### 4.1. The Severity of COVID-19 and One’s Responses

The severity of COVID-19 was positively associated with one’s behaviors, worries, and attitudes during the COVID-19 pandemic, indicating that the severer people think coronavirus is, the more adjustments they make. Concerning behaviors, people are more likely to wear masks and hoard things if they think coronavirus is more serious. Mask-wearing is a protective behavior that is recommended by the government, whereas hoarding is a coping strategy that is voluntarily adopted by individuals. Though both can help one to reduce anxiety and to restore a sense of control, hoarding is a relatively more negative coping strategy that can potentially spread anxiety in the society and harm the welfare of others. However, perceiving coronavirus as severer contributes more to hoarding compared to mask-wearing. This result implies to the authorities that finding a proper way to communicate the pandemic to the public is important in encouraging advocated deeds while discouraging unwanted ones.

Concerning worries, people who perceived coronavirus as more serious experienced more worries, and worried about contracting coronavirus themselves or spreading coronavirus to others. This might be because contracting COVID-19 is costly, not only in terms of potential severe sickness, such as death [37], but also in terms of reputational costs, such as being stigmatized by society [38].

Concerning attitudes, the severity of COVID-19 was associated with one’s discrimination and empathy toward those infected with coronavirus. The former implies that people who previously are infected with coronavirus and have recovered are likely to be labeled and stigmatized and may experience exclusion and unfair treatment. The latter suggests that people who currently have coronavirus are also likely to receive empathy from others. Noticeably, the severity of COVID-19 makes people more likely to feel empathetic toward those having coronavirus, relative to degrading those who once had coronavirus. The tendency for Chinese participants to exhibit more concerns and empathy, instead of blame and discrimination, toward those who are affected or have been infected previously by coronavirus might be due to the collective culture of China, as the Chinese are more concerned about the well-being of their groups [39]. Future research could examine the same issue in Western cultures to reveal potential cultural differences in this regard.

### 4.2. HEXACO Personality Traits and COVID-19 Related Responses

Concerning behaviors, Honesty-Humility and Conscientiousness were related to mask-wearing, indicating that people higher on these traits are more likely to take protective behaviors. The positive association between Honesty-Humility and mask-wearing echoes previous findings reporting a positive association between Honesty-Humility and various prosocial behaviors, such as cooperation in economic games [40], organizational citizenship behaviors [41], and ethical behaviors [42]. In the times of the COVID-19 pandemic, wearing masks could certainly be considered a prosocial behavior, regarding protecting not only oneself, but also others. Accordingly, people high in Honesty-Humility are more likely to enact this behavior. The positive correlation between Conscientiousness and mask-wearing can not only be interpreted by the concept of Conscientiousness, comprising normative and rule-following tendencies [34,43], but also by its consistent association with various health-promoting behaviors, such as exercising more [44], smoking less [45], and adhering to better medication [46]. Concerning hoarding, we only found it positively associated with Emotionality, which was also reported by Dammeyer [11]. This might be due to people higher in Emotionality tending to overreact under stressful situations [47]. Noticeably, we found no link between Honesty-Humility and hoarding despite such an association being reported by some previous studies [12], thus highlighting the importance of researching different samples.

Concerning COVID-19 related worries, we found that people higher in Emotionality reported more worries, worrying about contracting COVID-19 and spreading COVID-19. This conclusion echoes a recent meta-analysis linking emotionality to various worries related to COVID-19, including worrying about the threats that COVID-19 posed to their health, social life, close others, and society [34]. Also, Aschwanden et al. [6] found that people high in Neuroticism tended to be more concerned about COVID-19, such as fear of contracting the illness and becoming severely ill or dying from the disease. Besides Emotionality, we also found that Honesty-Humility, Extraversion, and Agreeableness were positively associated with worrying about spreading COVID-19 to others. The result that people higher in Honesty-Humility and Agreeableness worry more about spreading COVID-19 might be due to the altruistic nature of these traits, and thus people higher in these traits are more reluctant to harm or induce losses in others [48]. The association between Extraversion and worrying about spreading COVID-19 might be because people higher in Extraversion are afraid of risking their social life and/or being less popular among their social circles, which are the primary focus of extraverts [49]. Concerning COVID-19-related attitudes, almost all HEXACO traits were associated with these attitudes. Specifically, Honesty-Humility, Agreeableness and Openness to Experience were negatively correlated with discrimination toward people infected with coronavirus. This might be because people high in these traits prefer equality and social justice [50]. Accordingly, they are more lenient and less judgmental of individuals even though they are different from the majority in the sense of being infected with coronavirus. Emotionality was negatively correlated with discrimination toward people infected with coronavirus. Possibly, people higher in Emotionality tend to experience more intense worries and fears, driving them to place more blame on individuals infected with the coronavirus. People high in Conscientiousness exhibit higher discrimination toward individuals who have recovered from coronavirus. This might be because people high in Conscientiousness consider these individuals a threat to social order, which they prefer to have, thus putting more blame on them. Future studies should continue examining this.

All HEXACO traits were positively correlated with empathy toward people infected with coronavirus. Indeed, our results mirror previous findings that all the HEXACO traits are negatively correlated with at least one of the dark traits, i.e., Machiavellianism, Narcissism, Psychopathy, and Sadism [51,52]. As the core of these dark traits involves a lack of empathy, people who score high on the HEXACO traits are more likely to be concerned about the suffering of others. Noticeably, the three personality traits, Honesty-Humility, Emotionality, and Agreeableness, representing active altruism, reactive altruism, and kin altruism, respectively [48], have a relatively higher correlation with empathy toward people infected with coronavirus. This indicates that people with altruistic personality traits are more likely to care about individuals who are suffering.

### 4.3. Limitations and Future Directions

This study has some limitations. The reliability of BHI is relatively low in our sample. Still, most of the reliabilities of the sub-scales are comparable to that of the original [30] or studies using BHI [7]. Indeed, this is a common issue faced by short inventories, such as the 10-item Big Five Inventory (BFI-10) [53], as the Cronbach alphas tend to underestimate the reliability of heterogeneous scales [54]. Also, our study has more females than males, preventing us from generalizing our results. Future research should use samples with a more balanced gender proportion to examine this topic. Besides this, our study primarily focused on examining the associations between factors influencing one’s responses during the COVID-19 pandemic, and so future research should continue exploring mechanisms to provide explanations for these links.

## Figures and Tables

**Table 1 behavsci-14-00482-t001:** Descriptive statistics of study variables.

Variable	*M* (*SD*)	1	2	3	4	5	6	7	8	9	10	11	12	13	14
1. Honesty-Humility	3.83 (0.69)	0.43													
2. Emotionality	3.44 (0.63)	0.05	0.23												
3. Extraversion	3.62 (0.71)	−0.05	−0.05	0.50											
4. Agreeableness	3.10 (0.62)	0.23 **	−0.11 **	−0.01	0.20										
5. Conscientiousness	3.66 (0.74)	0.13 **	−0.14 **	0.12 **	0.08 *	0.59									
6. Openness to Experience	3.62 (0.66)	−0.15 **	−0.02	0.20 **	−0.01	0.11 **	0.34								
7. Mask Wearing	6.69 (0.82)	0.13 **	0.01	0.06	0.08 *	0.14 **	0.03	-							
8. Hoarding	4.03 (1.89)	−0.04	0.08 *	−0.02	0.03	0.02	0.02	0.14 **	-						
9. Infect	4.21 (1.17)	−0.03	0.18 **	−0.02	−0.02	−0.04	−0.01	0.13 **	0.24 **	0.64					
10. Spread	6.26 (1.07)	0.12 **	0.07 *	0.08 *	0.10 **	0.04	0.00	0.20 **	0.12 **	0.31 **	0.59				
11. Discrimination	2.50 (1.11)	−0.05	0.08 *	0.02	−0.14 **	0.09 **	−0.09 **	0.02	0.13 **	0.19 **	0.06	0.69			
12. Empathy	5.72 (1.15)	0.17 **	0.08 *	0.14 **	0.16 **	0.12 **	0.08 *	0.24 **	0.22 **	0.23 **	0.31 **	−0.01	0.58		
13. Severity	5.51 (1.01)	0.01	0.05	−0.03	0.04	0.03	−0.03	0.18 **	0.20 **	0.44 **	0.28 **	0.10 **	0.26 **	0.65	
14. Age	30.79 (10.46)	0.23 **	−0.11 **	−0.01	0.02	0.16 **	−0.13 **	0.01	−0.09 **	−0.05	−0.02	0.19 **	0.12 **	0.04	-
15. Gender	1.76 (0.43)	0.15 **	0.29 **	0.05	0.05	−0.09 **	−0.06	0.08 *	0.06	0.03	0.03	0.07 *	0.04	−0.04	−0.05

Note. Reliabilities are presented in the diagonal. For the coding of gender, male = 1, female = 2. * *p* < 0.05, ** *p* < 0.01.

**Table 2 behavsci-14-00482-t002:** Predicting COVID-19-related responses from the HEXACO traits and the severity of COVID-19.

	Variables	Behaviors				Worries				Attitudes			
		Mask Wearing		Hoarding		Infect		Spread		Discrimination		Empathy	
		ΔR^2^	β	ΔR^2^	β	ΔR^2^	β	ΔR^2^	β	ΔR^2^	β	ΔR^2^	β
**Step 1**		0.01		0.01 **		0		0		0.04 **	0.19 **	0.02 **	
	Age		0.01		−0.16 **		−0.05		−0.02		0.08 *		0.12 **
	Gender		0.07 *		0.10		0.03		0.03				0.05
**Step 2**		0.03 **		0.01		0.03 **		0.04 **		0.05 **		0.07 **	
	Honesty-Humility		0.09 **		−0.09		−0.04		0.11 **		−0.11 **		0.12 **
	Emotionality		0		0.13 *		0.18 **		0.08 *		0.08 *		0.12 **
	Extraversion		0.03		−0.05		−0.01		0.08 *		0.02		0.12 **
	Agreeableness		0.03		0.08		0.01		0.09 *		−0.12 **		0.14 **
	Conscientiousness		0.10 **		0.10		0		0.02		0.11 **		0.07 *
	Openness to Experience		0.02		0.02		−0.01		0		−0.10 **		0.08 *
**Step 3**		0.03 **		0.04 **		0.20 **		0.07 **		0.01 **		0.07 **	
	Severity		0.15 **		0.37 **		0.44 **		0.28 **		0.09 **		0.25 **

Note. * *p* < 0.05, ** *p* < 0.01.

## Data Availability

The data and analytical syntax of this study are available at: https://osf.io/cy4bs/?view_only=d7e5a9ddf38d42a8ab5083115d98b97d (accessed on 25 May 2024).

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
