# Peer review of "Understanding COVID-19-Related Behaviors, Worries, and Attitudes among Chinese: Roles of Personality and Severity"

_behavsci, 2024, doi:10.3390/bs14060482_

Round 1

Reviewer 1 Report

Comments and Suggestions for Authors

Review of the article: Understanding COVID-19 related behaviors, worries, and 2

attitudes among the Chinese: roles of personality and severity, Behavioral Science

The article examined how situational factors (i.e., the severity of COVID-19) and individual differences (i.e., the HEXACO traits) affect one’s COVID-19 related responses regarding behaviors (i.e., mask-wearing and hoarding), worries (i.e., worrying about infecting and spreading COVID-19) and attitudes (i.e., discrimination and empathy toward people infecting COVID-19) in China.

The article discusses important issues, but it has several weaknesses in terms of theory, methodology, and statistics. The introduction lacks theoretical justification for the selection of variables for analysis. The article only explains situational variables (Covid's severity) and individual variables (related to personality traits). The author's understanding of worries and attitudes, particularly empathy and discrimination towards people infected with COVID-19, is not addressed in the research. The research hypotheses were not stated. Table 1 presents both descriptive statistics and correlations, but it is unclear which correlation coefficient was used. It is also unclear whether the assumptions for the regression analysis were met and what the reliability indices of the research tools used are. The discussion of the results was presented objectively, with references to various research findings. With revisions and re-review, I believe the article can be published.

Reviewer 2 Report

Comments and Suggestions for Authors

First, thank you for the opportunity to review this work.

The study examined the effect of situational factors (e.g. the impact of COVID-19) and individual differences (e.g. HEXACO personality traits) and responses under changing and uncertain conditions and situations (e.g. the pandemic caused by COVID-19) in China. 

I find the study very interesting; I invite the authors to implement some minor aspects. 

The abstract is clear and well structured, even the keywords. 

I suggest adding the keyword COVID-19 to facilitate future research, as it is a central element of the work. 

The introduction and method are well described, as are the results. I recommend that the authors implement the discussions by starting as they describe (pp. 5- 6) and arguing their case by better clarifying the gap that enhances the results obtained in their work. 

Another imprinting element concerns the work's implications; it would be appropriate to describe in more detail what kind of contribution the study wants to make to the scientific community and beyond.  

The authors should include further references to understand better and discuss the variables described. 

Round 2

Reviewer 1 Report

Comments and Suggestions for Authors

Thank you for responding to my comments and suggestions.  Following the corrections made by the authors, I recommend the article for publication